# The Ubiquitin-like Proteins of *Saccharomyces cerevisiae*

**DOI:** 10.3390/biom13050734

**Published:** 2023-04-24

**Authors:** Swarnab Sengupta, Elah Pick

**Affiliations:** 1Department of Evolutionary and Environmental Biology, Faculty of Natural Sciences, University of Haifa Mount Carmel, Haifa 3498838, Israel; ssengupta@sci.haifa.ac.il; 2Department of Biology and Environment, Faculty of Natural Sciences, University of Haifa, Oranim, Tivon 3600600, Israel

**Keywords:** Atg8, Atg12, Hub1, NEDDylation, proteostasis, Rub1, *S. cerevisiae*, Smt3, SUMOylation, ubiquitin, Ubls, Urm1

## Abstract

In this review, we present a comprehensive list of the ubiquitin-like modifiers (Ubls) of *Saccharomyces cerevisiae*, a common model organism used to study fundamental cellular processes that are conserved in complex multicellular organisms, such as humans. Ubls are a family of proteins that share structural relationships with ubiquitin, and which modify target proteins and lipids. These modifiers are processed, activated and conjugated to substrates by cognate enzymatic cascades. The attachment of substrates to Ubls alters the various properties of these substrates, such as function, interaction with the environment or turnover, and accordingly regulate key cellular processes, including DNA damage, cell cycle progression, metabolism, stress response, cellular differentiation, and protein homeostasis. Thus, it is not surprising that Ubls serve as tools to study the underlying mechanism involved in cellular health. We summarize current knowledge on the activity and mechanism of action of the *S. cerevisiae* Rub1, Smt3, Atg8, Atg12, Urm1 and Hub1 modifiers, all of which are highly conserved in organisms from yeast to humans.

## 1. Introduction

Following their synthesis by ribosomes, many proteins undergo alterations by the covalent attachment to chemicals, lipids, carbohydrates, or even to other proteins. Such modulations are collectively termed “post-translational modifications” (PTMs) [1]. PTMs increase proteome diversity by manipulating protein properties, including their localization, protein–protein interactions, enzymatic activity, and/or turnover. The nature of PTMs can be either irreversible or reversible. Irreversible PTMs are mainly caused by chemical processes (such as oxidation, glycation, or deamidation) and lead to the formation of end products, whereas reversible modifications are transient and hence dynamic [2]. Transient PTMs are predominantly enzymatically added and removed, such that the different states of the modified protein can act as molecular on/off switches, enabling the rapid regulation of cellular signaling pathways or appropriate responses and adaptation to changing or unstable environments [1,3]. Examples of reversible modifications include methylation, acetylation, phosphorylation, and the attachment to members of the ubiquitin-like protein (Ubl) family to target proteins, but also to phospholipids. Transient post-translational modification of proteins by Ubls plays essential roles in human health and disease, with ubiquitin being the most studied family member [1,4]. Ubiquitin is a widely conserved 76-amino-acid-long polypeptide which shows 96% identity between the *Saccharomyces cerevisiae* and *Homo sapiens* orthologues. Ubiquitin modifies thousands of targets in processes that involve hundreds of enzymes, including those required for precursor maturation and subsequent covalent attachment to and release from substrates [5,6]. One of the most studied roles of ubiquitination is its pivotal function in the two key degradation machineries, namely, the ubiquitin-proteasome system (UPS) and lysosomal autophagy. Ubiquitination of substrates could be as a single moiety or as a polymeric chain. Polyubiquitination is primarily based on isopeptide linkages between specific Lys residues of ubiquitin and the C-terminal Gly residue of the next ubiquitin. Notably, Lys48-linked chains are the most common signal for proteasomal degradation and Lys63-linked chains are abundant upon stress [7,8,9]. It is therefore not surprising that ubiquitin is a critical regulator of various cellular mechanisms, including DNA damage and repair, cell cycle progression, apoptosis, and cell survival.

The family of Ubl modifiers shares structural properties with ubiquitin, including the well-defined ubiquitin fold, consisting of five β-sheets arranged around a central helix (Figure 1A). In terms of their mode of action, Ubl modifiers can be considered either canonical (such as *S. cerevisiae* SUMO/Smt3, NEDD8/Rub1 and Atg8) or non-canonical (such as *S. cerevisiae* Hub1, Atg12 and Urm1) (Table 1, Figure 1A,B). 

**Table 1 biomolecules-13-00734-t001:** *S. cerevisiae* ubiquitin-like paralogous proteins.

S. cerevisiae Ubls	E1	E2	E3	Substrate	References
Rub1 (YDR139C)	A dimer of Ula1(YPL003W) Uba3 (YPR066W)	Ubc12 (YLR306W)	Dcn1 (YLR128W) Hrt1 (YOL133W) Rad16 (YBR114W)	Cdc53 (YDL132W) yCul3 (YGR003W) Rtt101 (YJL047C)	[10,11,12]
Smt3 (YDR510W)	A dimer of Aos1 (YPR180W) Uba2 (YDR390C)	Ubc9 (YDL064W)	Siz1(YDR409W), Siz2/Nfi1 (YOR156C), Mms21/Nse2 (YEL019C) Slx5(YDL013W)/ Slx8(YER116C)	Tfg1 (YGR186W) PCNA(YBR088C) Cse4(YKL049C) Yku70(YMR284W)	[13,14,15,16,17,18,19]
Atg8 (YBL078C)	Atg7 (YHR171W)	Atg3 (YNR007C)	A complex of Atg12 (YBR217W) Atg5 (YPL149W) Atg16(YMR159C)	PE *	[20,21,22,23]
Atg12 (YBR217W)	Atg7 (YHR171W)	Atg10 (YLL042C)	—	Atg5 (YPL149W)	[24,25]
Urm1 (YIL008W)	Uba4 (YHR111W)	—	—	tRNA ** Ahp1 (YLR109W)	[26,27,28]
Hub1 (YNR032C-A)	—	—	—	Snu66 (YOR308C)	[29,30]

* PE—Phosphatidylethanolamine; ** tRNA thiolation and not URMylation.

Canonical Ubls are mostly translated as precursor proteins that must first undergo maturation by a Ubl-hydrolyzing enzyme that exposes the C-terminal Gly [31,32,33,34]. Attachment of the mature Ubls to their targets requires cognate cascades of E1-E3 enzymes (Figure 1C). Each cascade begins with hydrolysis of ATP and adenylation of the Ubl C-terminal Gly, which allows subsequent thioester formation between the exposed Gly residue and the catalytic Cys of a Ubl-specific activating (E1) enzyme. Activated Ubls are then transthiolized from E1 to the catalytic thiol of an E2 conjugating enzyme and are finally transferred to a positively charged target protein ε-amino side chain in a step mediated by a substrate-specific E3 ligase. Ubl modifications trigger and/or regulate various signals before being hydrolyzed by specific proteases. Non-canonical Ubls share properties with canonical modifiers, but they show unique binding characters or cascade enzyme dependency [5,21,35].

In this review, Ubls from the budding yeast *Saccharomyces cerevisiae* are considered. Budding yeast has long been used as a preferred model organism, given the wide variety of genetic, molecular, and biochemical tools and techniques available for working with this organism when addressing fundamental cellular processes that are conserved in complex multicellular organisms, such as humans. Although we have collected the most recent information on yeast Ubls, the information nevertheless remains partial, and so future studies are needed to reveal the underlying mechanisms involved in the actions of Ubls.

**Figure 1 biomolecules-13-00734-f001:**
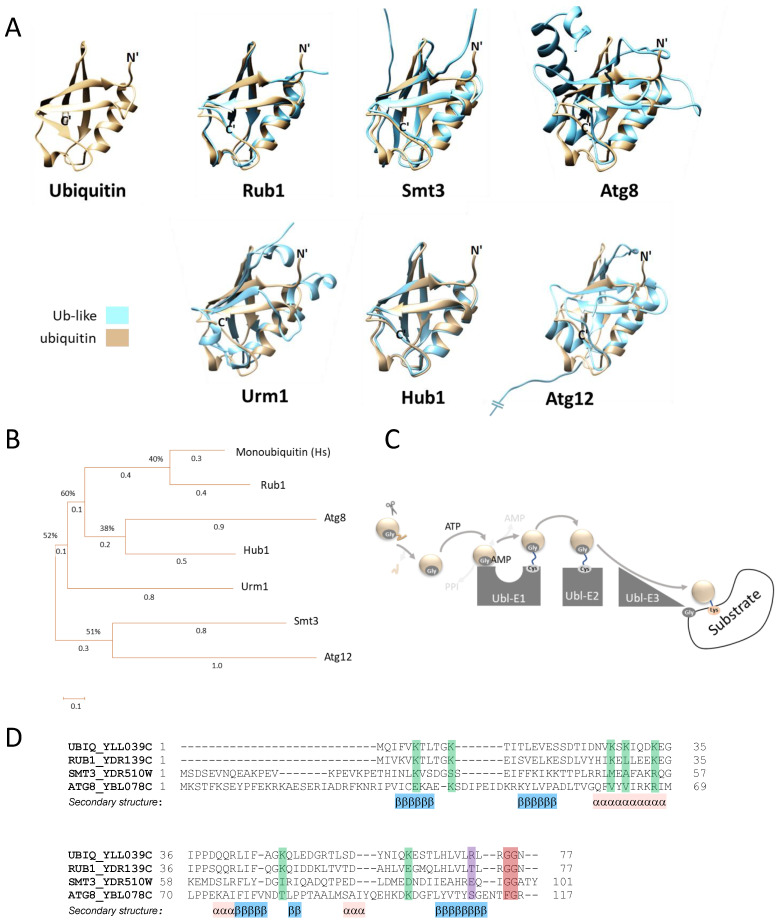
Ubiquitin and Ubls in *S. cerevisiae*. (**A**) Three-dimensional models of the *S. cerevisiae* ubiquitin-like modifiers Rub1, Smt3, Atg8, Atg12, Urm1 and Hub1. Protein structures were predicted by alphaFold [35], and superimposed with that of ubiquitin using the UCSF Chimera program (version 1.16). The amino and carboxy termini (N’ and C’) of ubiquitin are presented in black in each of the superimposed models. Note that the long amino terminus of Atg12 is not shown. (**B**) Phylogenetic tree of *H. sapiens* (Hs) ubiquitin and the six *S. cerevisiae* Ubls presented in A. Protein sequences were aligned with Clustal W software, and a neighbor-joining phylogenetic tree of the proteins was inferred from the multiple alignments using MEGA X. (**C**) Overview of the canonical ubiquitin/ubiquitin-like conjugation cascade. (**D**) Sequence alignment of *S. cerevisiae* ubiquitin and the canonical ubiquitin-like modifiers. Sequences were aligned with PROMALS3D server. The alignment indicates the various Lys residues that mediate ubiquitination (green), reveals the difference in charge of the residue used for E1 selection at position 72 (purple) and emphasizes the Gly residue used for conjugation (red).

## 2. Rub1 Is the Closest Paralog of Ubiquitin

Rub1 (Related ubiquitin 1) shares 59% identity with its human orthologue, NEDD8 (Neural Precursor Cell Expressed Developmentally Downregulated 8) [36]. NEDD8/Rub1 orthologues are essential for the viability of all studied model species, except for *S. cerevisiae* and its close relative *Candida albicans* [37]. Similarly to ubiquitin, Rub1 also includes 76 amino acids, 44 of which (57%) are identical to *S. cerevisiae* ubiquitin, with the total amino acid similarity being 77% (Figure 1B,D) [38]. Despite these prominent similarities, NEDD8/Rub1 and Ub employ parallel and cognate enzymatic cascades for post-translational modification of dissimilar substrates [39]. The NEDD8/Rub1-conjugating pathway (a.k.a NEDDylation) includes a set of E1/E2 enzymes and two E3s that work in concert [11,40]. To date, only four substrates are known to be modified by Rub1. Three of them belong to a single family of proteins known as “cullins” (Cdc53/yCul1, yCul3 and Rtt101/yCul4; see Table 1), each of which is used to produce a modular enzyme from the ubiquitin cullin-RING E3 ligase (CRL) family [41,42]. Interestingly, similarly to cullins, the fourth Rub1-modified protein is also a ubiquitin E3 ligase, namely, Rsp5, the only member of the NEDD4 family of HECT E3 ligases in *S. cerevisiae*. However, the general role of this modification remains unclear [43].

To modify its cullin substrates, the Rub1 precursor is first processed by Yuh1 (yeast ubiquitin C -terminal hydrolase 1), a protease that cleaves the C-terminal Asn residue to expose the di-Gly motif [42] (Figure 1D and Figure 2A). The covalent attachment of Rub1 to a specific Lys residue in one of the cullins is mediated by several enzymes, starting with a heterodimeric NEDD8/Rub1 E1 activating enzyme (NAE1). NAE1 comprises Ula1/Uba3 that adenylates (Ula1) and forms a thioester bond (Uba3) with the modifier. In fact, NAE1 specifically selects Rub1 over ubiquitin. Indeed, this E1 cannot activate ubiquitin due to a clash between two Arg residues, one within Ula1 and a second one at position 72 of ubiquitin [44,45,46]. However, because residue at the comparable position in Rub1 is not charged, it is sometimes activated by ubiquitin E1, albeit with lower efficiency [47]. In other organisms, this could lead to the NEDDylation of non-cullin substrates or to the formation of mixed ubiquitin-NEDD8 chains. However, this atypical NEDDylation is limited in *S. cerevisiae* only in the case of Rub1 over-expression or upon exposure to stress that depletes the levels of free ubiquitin [48,49,50]. Nevertheless, although the accumulation of Rub conjugates has been reported, no *S. cerevisiae* substrates have been identified to date.

From NAE1, Rub1 is transthiolized to the catalytic Cys residue of the E2 conjugating enzyme Ubc12. Following the formation of Ubc12~Rub1 thioesters, Rub1 is transferred to cullin substrates with the assistance of two E3s, Hrt1 (a.k.a. Rbx1) and the RING E3 subunit of CRLs, which cooperate with the co-E3 Dcn1 to stimulate cullin modification [40] (Figure 2A,B). The interaction between Ubc12 and Dcn1 is directed through N-acetylation of this E2, which brings the catalytic Cys of Ubc12 into close proximity with the cullin target, allowing Hrt1 to promote Rub1 transfer from Ubc12 to the cullin [11]. Interestingly, Hrt1 is an E3 ligase for both ubiquitin and Rub1, interacting with Ubc12 for NEDDylation and Cdc34 for ubiquitination. Indeed, the modularity of CRLs enable the formation of multiple and distinct E3 ligases, each of which modifies a cullin protein that is N-terminally attached to a diverged and changeable substrate receptor module and C-terminally attached to Hrt1 [12,51,52]. 

Of the three yeast cullins, only yCul1, which forms CRL1/SCF complexes, is vital. The substrate recognition module of the SCF (Skp, cullin, F-box) complex comprises a single adaptor, Skp1 (S-phase kinase-associated protein 1) that binds to various F-box substrate-specific receptors [51,53]. *S. cerevisiae* includes 18 proteins with F-box motifs, ten of which are confirmed SCF substrate receptors (i.e., Cdc4, Dia2, GRR1, Mdm30, Met30, Pfu1, Rcy1, Saf1, Ucc1, and Ufo1), five are putatively genuine F-box proteins due to their interactions with SCF components (i.e., Das1, Hrt3, Lug1, Skp2 and Ydr131C), and three have not yet been characterized (i.e., Roy1, Ctf13 and Mfb1). Among substrate receptors, only *CDC4* is essential for viability, with many of its conditional mutants resulting in cell cycle arrest at both G1/S and G2/M phases or exhibiting higher rates of chromosome loss at semi-permissive temperatures. Cdc4 recognizes and targets phosphorylated forms of key cell cycle regulators, such as Sic1, Cln1, Cln 2, Far1, and Clb6, for ubiquitination [52,54,55,56].

A less-studied *S. cerevisiae* CRL contains yCul3 as a scaffold protein. In metazoans, Cul3 interacts with BTB domain-containing proteins, which possess both adaptor and substrate receptor properties [52]. *S. cerevisiae* yCul3 forms at least two complexes, only one of which contains Hrt1 (i.e., yCul3-Hrt1-Elc1-Ela1) (Figure 2B) [54]. yCul3-Ela1 (the homologue of human Elongin A) is required for the poly-ubiquitination and proteasomal degradation of RNA polymerase II in response to DNA damage-induced stress. Interestingly, this activity first requires cleavage of Def1 (Degradation Factor 1) by the proteasome. The truncated Def1 is then translocated into the nucleus where it recognizes the largest subunit of RNA polymerase II (Rpb1) and targets it to the yCul3-Hrt1-Elc1-Ela1 complex for ubiquitination, followed by degradation [55,57]. The second yCul3-containing CRL includes Elc1 as an adaptor and Rad7, which is predicted to contain SOCS-box features characteristic of proteins involved in the assembly of CRL3 complexes, as a substrate receptor [56]. Indeed, mutations in this SOCS-box region abolish the ubiquitin E3 ligase activity of the complex. Uniquely, instead of the canonical Hrt1, this protein complex harbors the RING-containing protein Rad16 as an E3 subunit (Figure 2B). The yCul3-Elc1-Rad7-Rad16 complex targets the DNA damage-associated transcriptional repressor Rad4 for proteasomal degradation in response to UV radiation [58]. Notably, E2s involved in yCul3-Elc1-Rad7-Rad16 complex activity have not yet been identified.

Another yeast cullin is Rtt101/yCul4, which forms a CRL with Hrt1 and the adaptor protein Mms1 [12]. Similarly to Skp1, Mms1 recruits multiple substrate receptors, namely, Mms22, Crt10, Esc2 and Orc5 (Figure 2B) [59,60]. Rtt101-Mms1 complexes are involved in the regulation of chromatin metabolism, including replication initiation, replication fork stabilization and the regulation of mitotic nuclear division [12,59,61,62,63]. The Rtt101-Hrt1-Mms1-Mms22 complex is formed in response to DNA damage, whereas the Rtt101-Hrt1-Mms1-Crt10 complex has been suggested to function during replication by regulating nucleotide levels. The Rtt101-Hrt1-Mms1-Esc2 complex is involved in replication protection and the maintenance of genome stability. The role of the Rtt101-Hrt1-Mms1-Orc5 (subunit of the origin recognition complex (ORC)) has yet to be determined.

Cullin modification by Rub1 is reversed by the COP9 signalosome (CSN), which is a multi-subunit cullin-specific deNEDDylase (Figure 2A). The CSN harbors a JAMM/MPN+ metalloprotease motif within the catalytic Csn5 subunit, which hydrolyzes Rub1 from cullin [64,65]. The CSN shows high paralogy to the 19S proteasome lid, which contributes paralogous metalloprotease deubiquitination activity to the 26S proteasome [66,67]. Yet, unlike the high conservation of proteasome lid components across all eukaryotic phyla, the *S. cerevisiae* CSN shows more diversity, lacks subunits, and is not required for viability [68,69,70,71].

## 3. Smt3 Is a Vital Ubl

Smt3 is the *S. cerevisiae* orthologue of human SUMO-1. Smt3 was firstly revealed in 1995 in a screen for MIF2 suppressors [72]. Both the human and yeast orthologues were sequenced a year later [73]. Unlike Rub1, Smt3 and its E1/E2 enzymes are required for *S. cerevisiae* viability [74,75,76,77]. Smt3 is 22 amino acids longer than Ub and Rub1, and as a canonical Ubl, it harbors multiple Lys residues, as well as a carboxylic terminal Gly (Figure 1D). A deletion mutant lacking Smt3 can be complemented by the human orthologue, suggesting yeast-to-human conservation of this modifier [78]. The Smt3 modification cascade begins with maturation of its precursor through hydrolysis of a C-terminal Ala-Thr-Tyr extension by Ulp1 (Ubiquitin-like protease 1), a protease that specifically cleaves Smt3 extensions and chains (Figure 3). Following maturation, the modifier is transferred to a single heterodimeric E1 enzyme, comprising Aos1 and Uba2. Aos1 activates Smt3 through adenylation of its C-terminal Gly residue that next forms a thioester link with the catalytic Cys of Uba2 [79]. Smt3 is next transthiolized with its transfer from Uba2 to the catalytic Cys of the E2 enzyme, Ubc9. Lastly, with the aid of E3 enzymes, the modifier is transferred and covalently attached to its substrates, a.k.a. SUMOylation [14,79]. Notably, all SUMO E3 ligases identified so far are members of the PIAS (protein inhibitor of activated STAT) family of proteins. Each of these E3s harbors the SP-RING domain (in *S. cerevisiae*, these are Siz1/Siz2, Mms21, Slx5/Slx8, and Zip3), which interacts with Ubc9 to facilitate conjugation to substrates (Figure 3) [80,81].

Although Smt3 and its E1/E2 enzymes are vital, the only essential Smt3 E3 ligase in *S. cerevisiae* is Mms21. It was suggested that SUMOylation of Mms21 substrates are essential for viability. However, expression of a *mms21-RINGΔ* mutant, which does not harbor SUMOylation activity, is viable, although it exhibits strong phenotypes, suggesting that Mms21 vitality is distinct from SUMOylation [82]. The fact that none of the Smt3 ligases is essential for *S. cerevisiae* survival indicates either overlap in the specificity of these E3s for their substrates or that, under certain circumstances, Ubc9 SUMOylates substrates in vivo without recruiting E3 ligases [82]. The modification of proteins by Smt3 is enhanced during the cell cycle and under various stresses. In addition, Smt3 is also associated with transcriptionally active genes that regulate chromatid cohesion, chromosome segregation, DNA replication and septin ring dynamics [76]. In addition, it was recently shown that Smt3 regulates the quality control of misfolded and aggregated proteins via SUMOylation [83].

## 4. Atg8 and Atg12: Two Ubls in the Same Pathway

The autophagy (Atg) recycling pathway [84] is conserved in all eukaryotes, including S. *cerevisiae.* The *S. cerevisiae* Atg system includes more than 30 components, among them being Atg8 (a.k.a. LC3) and Atg12, both of which harbor Ubl properties and structural homology [24,85]. Atg8 is synthesized as a 117-amino-acid-long precursor that requires Atg4, the only essential yeast protease involved in autophagy, for maturation and exposure of the C-terminal Gly (Figure 1D) [86]. On the other hand, Atg12 naturally ends with a Gly and thus does not need a maturation processing step (Figure 4A).

Atg8 and Atg12 are activated by Atg7, a common E1 enzyme that uses ATP to adenylate the C-terminal Gly of each modifier. In the next step, the modifiers form a thioester bond with the catalytic Cys of Atg7 [87]. The paths of the two modifiers subsequently diverge when their E1 thioester-linked form is led to the catalytic Cys of E2 enzymes, namely, Atg3 for Atg8 and Atg10 for Atg12 (Figure 4B). The two pathways then converge in such a way that one of the modifiers (Atg12) catalyzes the modification pathway of the other modifier (Atg8). Following Atg10~Atg12 thioester formation, Atg12 modifies a specific lysyl ε-amino group of Atg5. Atg12-Atg5 conjugation is probably irreversible because no protease has been found that causes hydrolysis of this interaction [24]. It was suggested that the Atg12-Atg5 complex enhances the recruitment of additional components, specifically Atg16 [88], which together function as an E3 ligase for Atg8 conjugation (Figure 4). Of the complex components, Atg12-Atg5 harbors ligase activity and Atg16 is required for recruitment of Atg12-Atg5 with Atg8 [22,88]. The Atg12-Atg5-Atg16 E3 ligase differs from typical E3 ligases (such as HECT and RING E3 ligases) in that it does not include a cysteine motif or a RING finger. It is also non-conventional as it uniquely promotes the formation of an isopeptide bond between the C-terminal Gly of Atg8 and the amino group of phosphatidylethanolamine (PE), a phospholipid present in biological membranes, and not with proteins [22]. Notably, the E2 component Atg3 mediates membrane rearrangements that enrich the PE pool, thus enhancing the formation of Atg8- PE [89,90]. Atg8-PE conjugates contribute to the membrane tethering and hemifusion that results in the expansion of the autophagosomal membrane, which is a crucial step in autophagy [86]. Atg8-PE conjugation is reversed via a delipidation reaction catalyzed by Atg4. Thus, Atg4 is a bifunctional enzyme, which is required for C-terminal hydrolysis and delipidation.

## 5. Urm1 Is a bi-Functional Ubl

Urm1 is a non-canonical Ubl that shares weak amino acid similarity with ubiquitin and displays the characteristic ubiquitin fold structure, including the C-terminal di-Gly motif (Figure 1A and Figure 5A). Urm1 was found to be a bifunctional Ubl, integrating the functions of prokaryotic sulfur carrier proteins with those of eukaryotic Ubl modifiers. Urm1 was firstly reported by Furukawa and colleagues [91], who searched for relatives of bacterial sulfur carrier proteins. This revealed a polypeptide with amino acid similarity to the bacterial proteins MoaD (molybdopterin synthase sulfur carrier subunit) and ThiS (thiamin biosynthesis protein). Together with Urm1, the researchers also identified Uba4, which shows similarity to E1 activating enzymes, as well as noting the accumulation of high-molecular-weight Urm1 conjugates. These findings suggested that Urm1 is a Ubl, and Uba4 is its E1 activating enzyme [26,91]. Similar to canonical E1 enzymes, Uba4 harbors an adenylating domain, which is used to activate the C-terminal Gly of Urm1 [92]. At the same time, the other part of Uba4 includes a rhodanese homology domain (RHD) that catalyzes the formation of a C-terminal thiocarboxylate with Urm1 (Urm1-SH), instead of the classical thioester bond (Ubl^Gly~Cys^E2). Thiocarboxylated Urm1 functions as a sulfur carrier, which is required for chemical nucleotide modification at the wobble site in cytoplasmic tRNAs [93]. Additionally, Urm1 also functions as a protein modifier, binding a specific Lys on the peroxiredoxin Ahp1 in response to increased oxidative stress (Figure 5B, Table 1) [94]. Although *URM1* is not essential, deletion mutants of the modifier or its cascade enzymes exhibit increased temperature sensitivity, morphological alterations, and defects in protein homeostasis [28]. Interestingly, Urm1-deficient yeast cells present increased sensitivity to oxidative stress induced by tert-butyl hydroperoxide or diamide [92]. Unlike what occurs with canonical Ubls, the covalent attachment of Urm1-SH to Ahp1 in vitro is E2/E3-independent and requires oxidative stress. Furthermore, Urm1 leads to the persulfidation of cysteines in Ahp1, probably so as to protect these thiols from oxidative damage. Accordingly, it is thought that Urm1 acts as a stress response component under oxidative stress [28].

## 6. Hub1: A Ubl That Lacks a Carboxylic Terminal Glycine

Hub1 (Homologous to ubiquitin 1) is a 73-amino-acid-long non-essential UBL (a.k.a UBL5 in mammals). Similar to other Ubls, Hub1 also displays the characteristic ubiquitin fold structure [95] (Figure 1A). However, this unconventional Ubl lacks a canonical C-terminal Gly residue required for conjugation, and instead presents an evolutionarily conserved di-Tyr (YY) motif that is followed by the Leu73 residue in budding yeast (Figure 6A) [96]. It was previously shown that a mutant lacking Leu73 can produce high-molecular-weight conjugates [29]. To date, neither a specific protease for trimming Leu73 nor an enzymatic E1-E2-E3 cascade have been identified for this modifier.

Hub1 noncovalently binds components of the mRNA splicing factor U4/U6.U5 tri-snRNP (Snu66, Prp38 and Spp381) through a conserved HIND (Hub1-interacting domain) element presented by Snu66. The binding of Hub1 to U4/U6.U5 tri-snRNP has a slight effect on splicing but promotes the appearance of an alternative splice variant of SRC1 (Spliced mRNA and Cell cycle regulated gene 1) pre-mRNA, an important player for the stability of sister chromatid segregation [30,97]. This activity of Hub1 leads to the formation of Heh1-S, a short splice variant of SRC1 pre-mRNA, in addition to the regular mRNA Heh1-L (Figure 6B). Hub1 has been reported to play a regulatory role in nuclear envelope (NE) integrity by recruiting ESCRT-III (the endosomal sorting complexes required for transport 3) to NE via the ESCRT-III adaptor Chmp7. Capella et al. showed that mutants of ESCRT-III exhibited excessive recruitment of Chmp7 to the NE, causing membrane scission and severe cellular damage [98]. The damage in these mutants was suppressed by Hub1 expression, which became vital for survival. The essentiality of Hub1 resulted from induction of a Heh1-S splice variant, in addition to the canonical Heh1-L, which together form heterodimeric mRNA that is required to suppress cytotoxicity in ESCRT-III mutants [99,100,101]. Accordingly, it was suggested that the absence of Heh1-S leads to a toxic gain-of-function phenotype of the Heh1-L monomers, characterized by the defects described above.

A link between Hub1 and splicing was additionally suggested by its binding to Prp5, an evolutionarily conserved RNA helicase involved in splicing. The binding of Hub1 to Prp5 is not mediated through the HIND element but through a specific Trp residue that stimulates Prp5 ATPase activity and increases overall splicing efficiency [102]. Interestingly, Hub1 is essential for the viability of human cells by being involved in the optimal and correct splicing of a vast number of immature mRNAs [95,103]. Aside from splicing, the involvement of Hub1 in redox homeostasis was also suggested. Over-expression of Hub1 was reported under oxidative stress, hypo-osmotic pressure, and in the presence of heavy metals such as cadmium [30,95]. This stress-induced over-expression is transcriptionally regulated by YAP1 (Yeast AP-1), a basic leucine zipper (bZIP) transcription factor required for oxidative stress tolerance [95].

## 7. Summary and Prospective

*S. cerevisiae* is a widely used experimental model organism for which extensive tools to study basic cellular functions are available, including the biology of Ubls. The *S. cerevisiae* genome includes 11 proteins that show homology to ubiquitin, namely, the six Ubl modifiers addressed in this review and an additional five ubiquitin-like domain proteins (i.e., Ddi1, Ubp6, Mdy2, Rad23 and Dsk2), which contain several domains, one of which is a Ubl [104]. In this review, we focused on the biochemistry of Ubl modifiers, and their influences on cellular integrity via regulation of proteolysis through the 26S proteasome, regulation of autophagy, tRNA thiolation, nutrient sensing and regulation of pre-mRNA. These diverse biological processes contribute to the maintenance of cellular homeostasis.

The small number of Ubls and the fact that most (including their enzymatic cascades) are not required for viability has enabled researchers nowadays to focus on the cross-regulation between the conjugation pathways, the modification of proteins by multiple Ubls in an orchestrated manner to control biological signals, the use of the same Lys residue as a port for several Ubls, and the formation of mixed chains. For example, Rub1 can be incorporated into polyubiquitin chains, which in turn can be recognized by the proteasome [50]. It would be interesting to evaluate the power of such pathway crosstalk(s) in expanding cellular function and proteome diversity or improving the ability of an organism to fine-tune responses to alterations in the environment. Because these Ubl modifiers are relevant in health and disease, and as yeast has already been established as having a useful system for analyzing human diseases, additional knowledge on crosstalk and signaling could contribute to the development of novel therapeutic tactics.

## Figures and Tables

**Figure 2 biomolecules-13-00734-f002:**
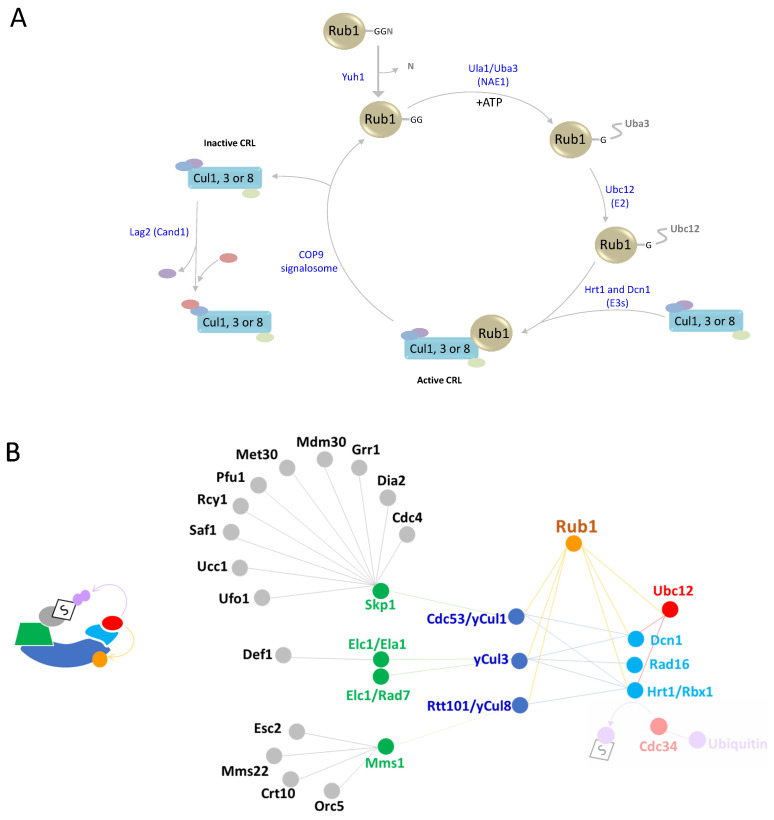
Rub1 cascade and properties. (**A**) Schematic depiction of the Rub1 modification cascade and cycle. The Rub1 precursor is processed by the carboxy terminal-acting hydrolase Yuh1. The carboxylic Gly residue of Rub1 forms an adenylate with NAE1, followed by thioester formation. Rub1 is then transthiolized to Ubc12. Finally, Rub1 is transferred to a Lys residue on one of three cullins by two E3 ligases, Hrt1 and Dcn1. Cullin NEDDylation is reversed by the COP9 signalosome, a cullin-specific DeNEDDylase. Notably, CRL components are recycled via the CAND1/Lag2 exchange factor in a mechanism that is beyond the scope of this review. (**B**) Interactions and functional relationship between CRL components and modifying enzymes, as detailed in the text.

**Figure 3 biomolecules-13-00734-f003:**
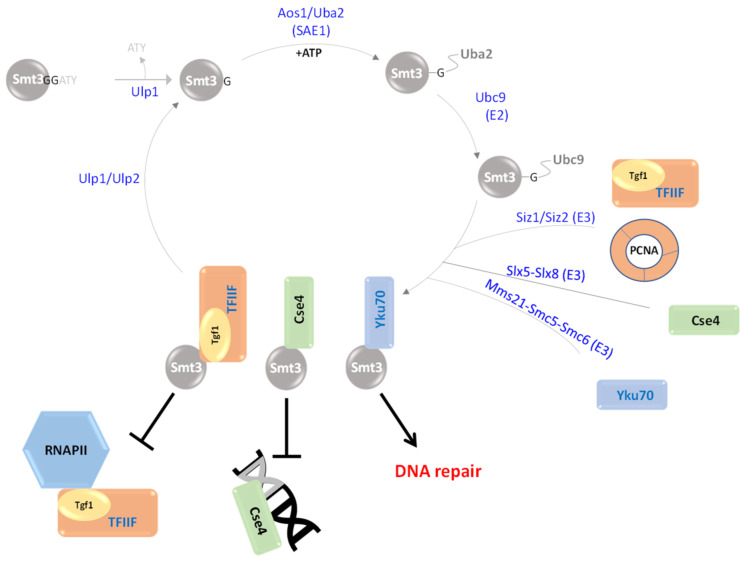
Schematic depiction of the Smt3 cascade. Following Smt3 maturation by Ulp1, the modifier is activated and forms a thioester bond with the heterodimeric Aos1/Uba2 E1 enzyme, followed by transthiolization to the E2 Ubc9. Smt3-Ubc9 interacts with various E3 ligases, resulting in the binding of specific substrates. The Siz1/Siz2 E3 ligase catalyzes the modification of Tgf1 (the large subunit of TFIIF) to avoid the pairing of RNA polymerase II with TFIIF. Siz1 mediates PCNA SUMOylation, and thus contributes to meiosis and genome stability. The E3 Slx5-Slx8 catalyzes Cse4 SUMOylation, which prevents its attraction to euchromatin under normal physiological conditions. The Mms21-Smc5-Smc6 complex facilitates SUMOylation of substrates such as the DNA repair component Yku70.

**Figure 4 biomolecules-13-00734-f004:**
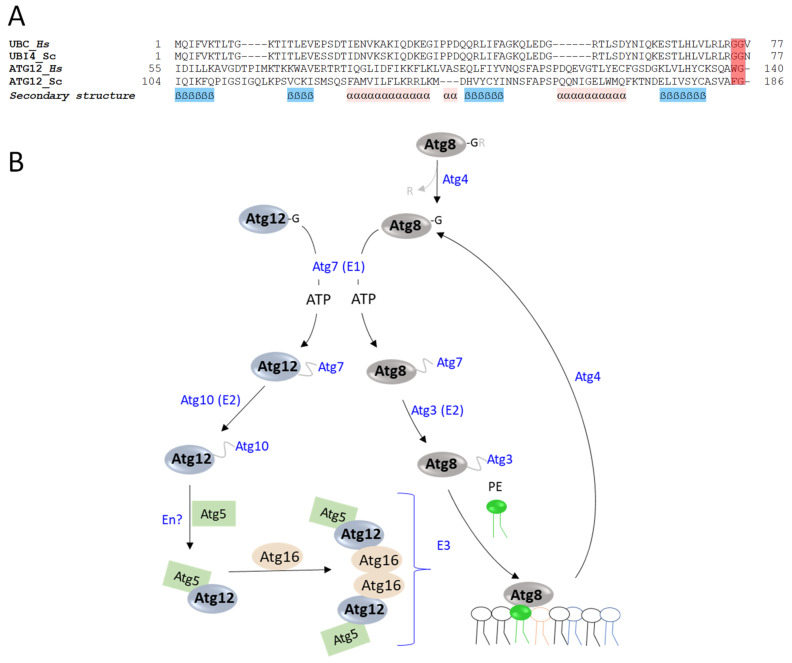
Atg8/12 cascades and properties. (**A**) Sequence alignment of ubiquitin (UBI4/UBC) and the carboxyl terminus of Atg12 from *H. sapiens* and *S. cerevisiae*. Sequences were aligned using the PROMALS3D server. (**B**) Schematic depiction of Atg8/Atg12 enzymatic cascades. Atg8 requires the Atg4 protease for its maturation. Subsequently, both Atg8 and Atg12 are activated by the E1-like enzyme Atg7, which first adenylates the C-terminal of each modifier following thioester formation. The next step includes transthiolation of the modifiers from Atg7 to cognate E2 enzymes, namely, Atg10 for Atg12 and Atg3 for Atg8. Next, Atg12 modifies Atg5 that in turn forms a complex with Atg16, all three of which act together as an E3 ligase that catalyzes the Atg8-PE conjugation.

**Figure 5 biomolecules-13-00734-f005:**
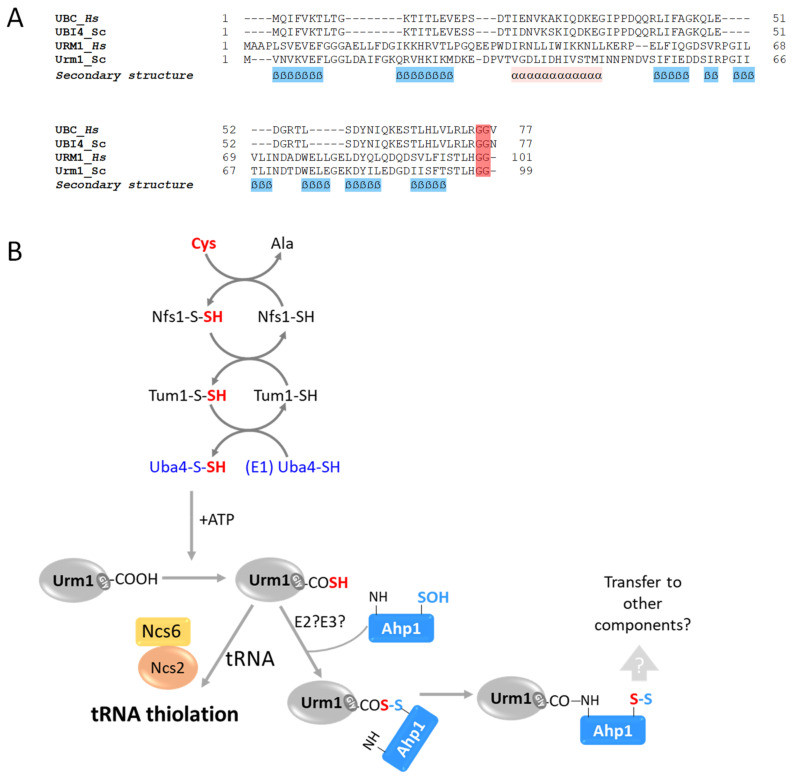
Urm1 cascade and properties. (**A**) Sequence alignment of ubiquitin (UBI4/UBC) and Urm1 from *H. sapiens* and *S. cerevisiae*. Sequences were aligned using the PROMALS3D server. (**B**) Schematic depiction of the Urm1 cascade. Uba4 is activated via the cascade of sulfur flow starting from cysteine (Cys) through the Nfs and Tum1 proteins. The activated form of Uba4 (Uba4-S-SH) acts as the E1 conjugating enzyme for Urm1 to generate subsequent modifications of Urm1, yielding either to Urm1-COSH or Urm1-CO-Uba4. Urm1-COSH is responsible for tRNA thiolation through Ncs6 and Ncs2 or for URMylation of targets such as Ahp1. The E2 and E3 of this process are still unknown.

**Figure 6 biomolecules-13-00734-f006:**
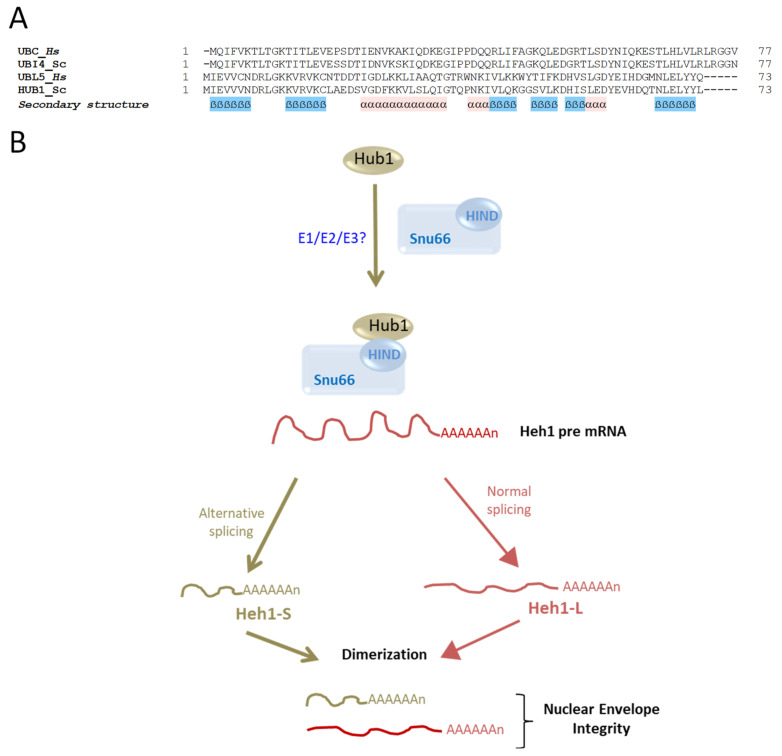
Hub1 cascade and properties. (**A**) Sequence alignment of ubiquitin (UBI4/UBC) and Hub1/UBL5 from *H. sapiens* and *S. cerevisiae*. Sequences were aligned using the PROMALS3D server. (**B**) Schematic depiction of the Hub1 cascade. Hub1 binds to a HIND motif present in the spliceosomal protein Snu66 and promotes the formation of Heh1-S, an alternative splicing of SRC1 pre-mRNA. This variant is dimerized with the canonical mRNA Heh1-L, and together they play an important role in the maintenance of nuclear envelope integrity.

## Data Availability

Not applicable.

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
