# Peer review of "The Ubiquitin-like Proteins of Saccharomyces cerevisiae"

_biomolecules, 2023, doi:10.3390/biom13050734_

Round 1

Reviewer 1 Report

The Ubiquitin-like modifiers (Ubls) family members are proteins displaying structural similarities with Ubiquitin (Ub). Ubls target both proteins and lipids. Similarly to Ub, the Ubls are processed, activated and conjugated to different substrates. Notably, as it occurs for Ub the conjugation process is reversible, meaning that once attached to the substrate/s Ubls can be enzymatically removed. Different substrates, regulating key cellular processes including cell-cycle progression, cellular differentiation and stress responses, can be regulated by Ubls. In their manuscript titled "The Saccharomyces cerevisiae ubiquitin-like proteins" Sengupta and Pick present an updated review on the budding yeast Ubls. The review is comprehensive and clear. Nonetheless, prior to publication minor concerns, below shortly listed, need to be edited. The lower part of Figure 1A seems to be cut. Please provide the complete picture. Additionally, the legend indicates N' and C', but there is only N' in bthe 3D structure pictures, thus the panel should be implemented. References should be carefully checked because sometimes they do not seem to properly fit/match with the main text (e.g.line 90, sure that is not 38-39? it seems that 40 matches better, is it not?) Even if some Ubls family members display weak similarities in their primary structure with Ub (e.g. Urm1) why not attempt to implement figures 4, 5, and 6 with a protein alignment as it has been done for figures 2 and 3? A few typos should be edited (e.g. lines 160 and 207).

The English language is fine, just minor editing is required (e.g. lines 161, 218-219).

Author Response

We thank the reviewer for the comments. Please find our answers below each of the comments:

The lower part of Figure 1A seems to be cut. Please provide the complete picture.

Atg12 is a 186 amino acids long protein with a huge NTD. We now changed the superimposition and added to the fig legends “Note that due to the length of the Atg12 amino terminus, it is shown fragmented.”. In addition, we added alignment of the most conserved part of Atg12 (C-terminal) in Figure 4A.

Additionally, the legend indicates N' and C', but there is only N' in the 3D structure pictures, thus the panel should be implemented.

We are sorry that it was not clear. The C’ is there, but the figure was too dark. We hope that it is clearer now.

References should be carefully checked because sometimes they do not seem to properly fit/match with the main text (e.g.line 90, sure that is not 38-39? it seems that 40 matches better, is it not?).

The references were edited and reconsidered across the entire manuscript. We corrected the references in multiple places.

Even if some Ubls family members display weak similarities in their primary structure with Ub (e.g. Urm1) why not attempt to implement figures 4, 5, and 6 with a protein alignment as it has been done for figures 2 and 3?

Thank you. We added additional alignments. The canonical Ubls are now aligned in Figure 1D, and for non-canonical Ubls in Figures 4A, 5A and 6A.

A few typos should be edited (e.g. lines 160 and 207).

Thank you! The manuscript was heavily edited by a professional proofreader.

Reviewer 2 Report

This is a nicely-written review article on ubiquitin and ubiquitin-like proteins as part of the posttranslational modification system in eukaryotes. The article is well-illustrated with Figures, although some minor modifications as listed below might be beneficial for the readers. 

Minor points.

1. Table 1. Kindly align the Table elements to improve the presentation. The title line can have one line only. E.g., a) "orthologues" can be moved to the main text or Table description); b) "NEDD8" can be aligned with "Rub1"; c) "Hub1 (YNR032CA)" would also look nicer in one line (simply reduce the font?) I am not sure bold characters are accepted by the journal, but in any case, bold is not needed in this Table as well.

2. The paragraph (lines 54-75) is long. It could be split into two or reduced for more comfortable reading. The same for lines 243-264. And lines 274-299. 

3. Figure 1C uses very small fonts. Kindly consider increasing to make reading easier. Even some letters in Figure 1A and 1B are of a too small font as well. 

4. Figure 3A. There are symbols under the amino acid sequence (:.*). One can guess the meaning of the symbols, however, it would be beneficial to directly explain the meaning in the Figure or in the Figure legend and text, when mentioned in the text. Kindly check if the symbols are correctly placed. 

5. Line 278 "colleagues". Kindly check.

6. Overall, there are many typos, such as extra spaces and extra or missing dots. It needs to be proofread.

7. Line 314. While "nither" is an understandable word, isn't it more standard to use "neither" instead?

8. It seems that less than 20% of references are from the recent 5 years. For a review article, it is expected to prioritize original papers of all time, and mainly recent reviews, when needed.

There are many small typos (dots, spaces, spelling) which is easy to fix. The authors need to carefully proofread it.

Author Response

This is a nicely-written review article on ubiquitin and ubiquitin-like proteins as part of the posttranslational modification system in eukaryotes. The article is well-illustrated with Figures, although some minor modifications as listed below might be beneficial for the readers. 

We thank the reviewer for the very nice words.

Minor points.

  1. Table 1. Kindly align the Table elements to improve the presentation. The title line can have one line only. E.g., a) "orthologues" can be moved to the main text or Table description); b) "NEDD8" can be aligned with "Rub1"; c) "Hub1 (YNR032CA)" would also look nicer in one line (simply reduce the font?) I am not sure bold characters are accepted by the journal, but in any case, bold is not needed in this Table as well.

As per the suggestions, we deleted the “orthologues” column and organized the table according to the suggestion. As this reviewer commented, the final decision on table structure is up to the journal editor.

  1. The paragraph (lines 54-75) is long. It could be split into two or reduced for more comfortable reading. The same for lines 243-264. And lines 274-299. 

Thank you, done.

  1. Figure 1C uses very small fonts. Kindly consider increasing to make reading easier. Even some letters in Figure 1A and 1B are of a too small font as well. 

Thank you! We changed the resolution and size of the figure.  

  1. Figure 3A. There are symbols under the amino acid sequence (:.*). One can guess the meaning of the symbols, however, it would be beneficial to directly explain the meaning in the Figure or in the Figure legend and text, when mentioned in the text. Kindly check if the symbols are correctly placed. 

We changed the alignments and their graphics: The canonical Ubls are now aligned in Figure 1D, and for non-canonical Ubls in Figures 4A, 5A and 6A.

  1. Line 278 "colleagues". Kindly check.

Thank you. Done.

  1. Overall, there are many typos, such as extra spaces and extra or missing dots. It needs to be proofread.

Thank you.  According to this comment, the manuscript was sent to an external language editor for proofreading.

  1. Line 314. While "nither" is an understandable word, isn't it more standard to use "neither" instead?

Done.

  1. It seems that less than 20% of references are from the recent 5 years. For a review article, it is expected to prioritize original papers of all time, and mainly recent reviews, when needed.

We thank the reviewer for the comment. We reorganized the references, and 11 unnecessary review articles were eliminated. Yet, several old reviews are important due to the integration they suggested and accordingly, we decided to keep them.

There are many small typos (dots, spaces, spelling) which are easy to fix. The authors need to carefully proofread it.

Thank you.  As written above, the manuscript has been proofread. 

Reviewer 3 Report

In this manuscript the authors review the current knowledge concerning the reversible post-translational modification of proteins by ubiquitin-like proteins (Ulps). This manuscript is interesting and well written.

Minor comment:

Lines 99-102: please see Α. H. Megarioti et al.,2021. Int. J. Mol. Sci. 22, 10208 https://doi.org/10.3390/ and references therein.

Author Response

Thank you very much.